# The Legacy of Potential Environmental Soil Contamination in an Antimony Mining Heritage Area

**António Fiúza \*, Aurora Futuro, Joaquim Gois, M. Lurdes Dinis, Cristina Vila, Soeiro Carvalho and António Fernandes**

CERENA—Polo FEUP—Centre for Natural Resources and the Environment, Faculty of Engineering, University of Porto, 4200-465 Porto, Portugal

\* Correspondence: afiuza@fe.up.pt

**Abstract:** In the Valongo Belt, with an extension of about 90 km, located very close to Porto, northern Portugal, dozens of ore deposits of various metallic minerals and coal were exploited in the 19th and 20th centuries. One of the metals most intensely exploited was antimony, with or without associated gold mineralization. This research intends to verify the extent of the current environmental legacy of ancient antimony mining. A typical old mine was selected. The main objectives were to verify whether the environmental legacy still manifests today, how natural processes contributed to an environmental dispersion of the mining footprint and whether the environmental legacy was absorbed by developments in a new landscape modified by anthropic activities. The topography of the area was captured using a Light Detection and Ranging (LIDAR) based drone system. The regional background was characterized by the geo-referenced chemical analysis of 157 soil samples, collected in a 35 × 35 m grid. The former mining area was characterized by 58 supplementary samples. The mining area is distinct from the background by higher antimony and zinc levels, constituting two distinct populations, as confirmed by statistical tests. In the samples collected in the industrial zone, six elements were considered contaminants: As, Cu, Mo, Sb, Sn and Zn. The concentrations of these elements were statistically examined using multivariate statistical analyses (principal component analysis and correspondence analysis). The main conclusions are: (a) the mining heritage area is discernible from the highly mineralized background; (b) in the mining zone, it is possible to distinguish the processing industrial area from the waste rock storage; (c) the natural processes of environmental dispersion were of little relevance; (d) the environmental legacy was smoothed and mostly incorporated into the new post-industrial landscape created by anthropic activities.

**Keywords:** mining environmental heritage; soil contamination; Valongo Mineral Belt; principal component analysis; correspondence analysis

## 1. Introduction

The Valongo Belt extends in a range of about 90 km, with NW–SE orientation, between Esposende and Castro Daire, in the NW of Portugal (Figure 1). In this mineral belt was found the mineralization of gold–antimony, lead–zinc–silver and tin–tungsten. Many of them were exploited as deposits since the Roman occupation of the Iberian Peninsula. This mining district was very important, having been the second-largest gold producer in Portugal [1–3].

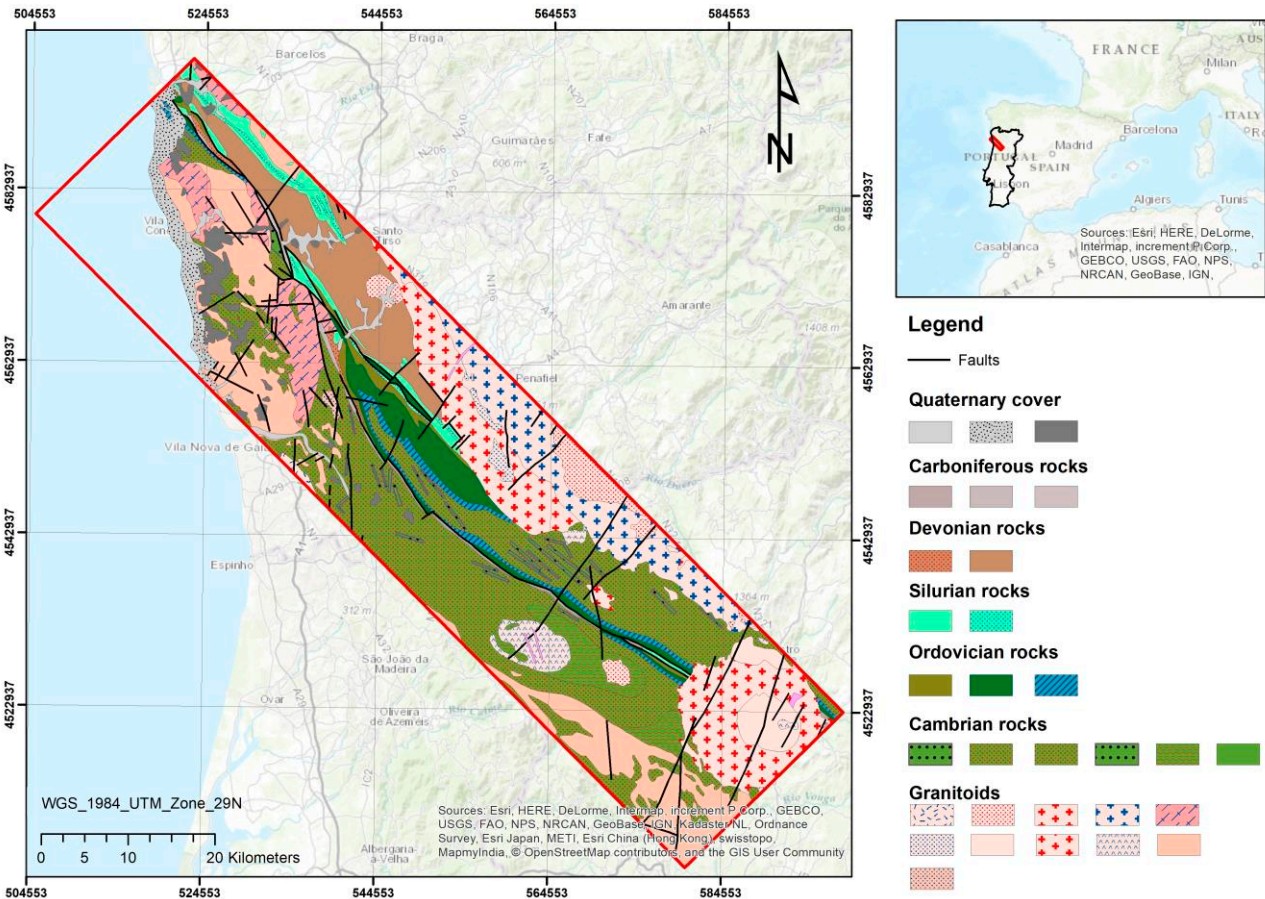

**Figure 1.** Valongo Mineral Belt [4,5].

The first exploitations of antimony, which often occur associated with gold, are much more recent. The first deposits were discovered in 1807, but they only became subjected to active mining in 1858. Exploitation peaked in the 1870s–1890s. The production of antimony ceased practically from the end of the 19th century. During the 20th century, it occurred in sporadic exploitations of minor amplitude, ceasing completely in 1971 [6].

The importance of the mineral belt of Valongo is well evidenced by the fact that the Map of Douro Mining Region (1884–1891) indicates the existence of 61 Sb Mines in the mineral belt of Valongo [7].

Antimony is a toxic metalloid that occurs in the earth's crust at very low concentrations between 0.2 and 0.6 ppm [8]. It has been known since pre-dynastic Egypt [9], where it was used as a cosmetic, as well as in Rome, as mentioned by Plinius Secundus. The first discovery of native antimony took place at the Sala Silver Mine in Sweden.

Nowadays, antimony has wide use, such as in alloys, flame-proofing compounds, batteries, plastics, ceramics, glass, detectors and diodes, cable sheathing, small arms, paints and medicine. It is considered a critical raw material by the European Union (EU).

At the end of the 19th century antimony was a metal widely exploited in Portugal, mainly in the Valongo Mineral Belt, where existed many dozens of active mines. During the 20th century exploitation became occasional and only in the most important mines. Parts of the locations of the former mines have new uses today, ranging from churches to supermarkets. However, most of these sites still exist in non-agricultural areas, sometimes forested, relatively inaccessible and of low population density. The mine wastes accumulated at those mine sites resulted from a very rudimentary ore treatment in relation to the processes that are applied today.

The mine selected for our research, the Ribeiro da Serra Mine, was one of the most important and processed ores of another small mine called Fontinha, also exploited

underground. At one point, the two underground exploitations were linked to each other. The largest production occurred in 1884 and was 1437 t of ore. The grades were high, normally above 5% Sb. The ore with grades 1%–2% Sb was considered low-grade, 5%–25% Sb was the average grade and ores with 45%–60% Sb were high-grade. The ore processing involved primary and secondary crushing followed by manual choice and roasting. Later, gravity concentration by shaking tables was also introduced. Intermediate temperatures between volatilization and fusion were used to roast the concentrates, producing "crudum" in crucibles.

While the environmental dispersion of As has been studied for decades, Sb has deserved little attention. Antimony mining always generates waste with As and Sb content [10].

Antimony is classified as a strong chalcophile, occurring with sulfur and some heavy metals [11]. It occurs in concentrations less than 0.3–8.4 mg kg$^{-1}$ in soils and less than 1 lg l$^{-1}$ in unpolluted water. It is commonly assumed that antimony is similar to arsenic in both chemical behavior and toxicity [11].

When released into the environment, Sb strongly attaches to particles that contain iron, manganese or aluminum [12,13]. Major sources of pollution during antimony (Sb) mining and processing are waste rock, tailings, smelting waste and wastewater in underground galleries [13].

Zhao et al. [14] made a global-scale assessment of Sb contamination in soil that was related to mining/smelting activities which was conducted based on 91 articles that were published between 1989 and 2021. In Portugal, high soil Sb concentrations were detected in the Sarzedas mine in the Castelo Branco County (663.1 mg/kg) (Pratas et al., 2005 [15]) and in the São Domingos copper sulfide mine in the Baixo Alentejo Province (467.8 mg/kg) (Anawar et al., 2013, [16]).

Antimony toxicity occurs either due to occupational exposure or during therapy. Occupational exposure may cause respiratory irritation, pneumoconiosis, antimony spots on the skin and gastrointestinal symptoms. In addition, antimony trioxide is possibly carcinogenic to humans [17].

In the historical period when Sb was exploited in Portugal, there were no environmental concerns. Meanwhile, many decades have passed since their effective abandonment without any human intervention. Some relevant questions may be raised:

– how the potential environmental impact of the mining activity still expresses;
– to what extent have natural processes contributed to an environmental dispersion in the areas with high background concentrations;
– to verify whether the environmental legacy has been nuancedly incorporated into a new spatial local reality from which it is already inseparable.

## 2. Site Description

Several criteria were considered to select the Sb-Au mine that would be the subject of study: the former importance of the mine, the accessibility, the existence of mining wastes, the proximity to streams and the representability of the site. Considering these criteria, the area of the former Ribeiro da Serra mine was selected. The topographical survey was performed using a drone with LIDAR technology given the actual intense forest cover of the site (Figure 2).

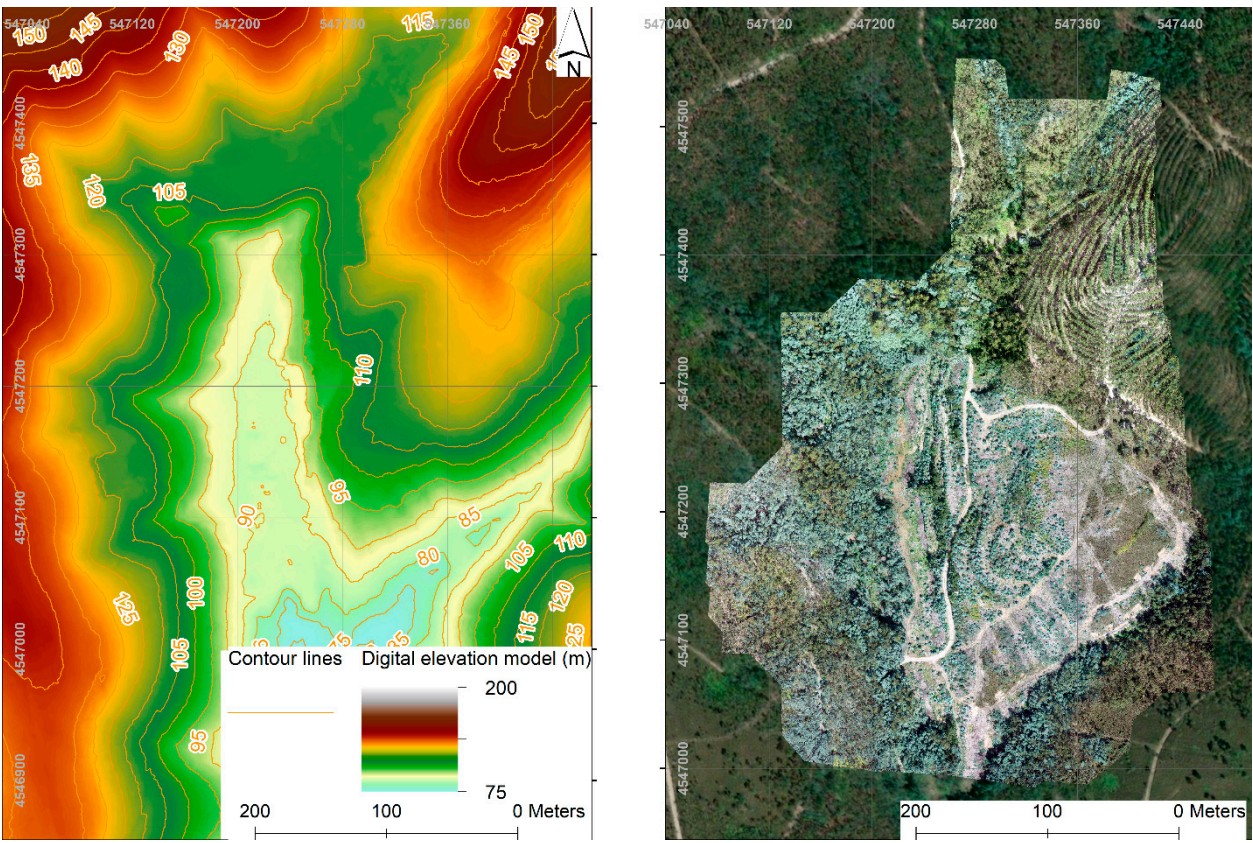

**Figure 2.** Topography survey o9f the site with a LIDAR-based drone system.

The former mine facilities were located on the right flank of a gentile valley where a water stream may seldom flow when rainfall is exceptionally hi3gh. The site is in northern Portugal—UTM (Universal Transverse Mercator) 29T, 547256.20 m E, 4547250.04 m N. Elevations rise in the area from 70 to 170 m high. The climate is temperate with mild seasonal temperatures. The average annual rainfall for the last 30 years was 1071 mm. The hottest days occur in July and August, with averages of 27 °C, while the coldest days occur in January with an average of 6 °C. The single hottest days occur in July and August, averaging 35 °C, while the coldest nights occur in January and December, averaging −1°C. The wind blows predominantly from ENE to WSW, with speeds between 12 and 19 km/h. The sea wind also blows regularly, from W to E, reaching the highest speeds between 28 and 39 km/h.

The area is predominantly occupied by eucalyptus crops mainly used for the paper pulp industry. Agriculture is sparse with very small fields mainly used for family consumption.

### 2.1. Geology and Mineralization

Recent metallogenetic interpretations clearly demonstrate that mineralization in the Valongo Belt was associated with the presence of ore fluids introduced by a series of faults and fractures, centered on veins, usually with massive zones of sulfides or quartz veins with distinct lithology. Antimony and gold both occur in granites as veins, stockworks and Paleozoic metasedimentary rocks, affected by different types of shears. The genesis is epigenetic and hydrothermal with paragenesis Sb-Au. Mineralization occurs in veins and quartz stockworks oriented N-S 20° to 40° W and E-W 20° to 65° N. The thicknesses of the veins vary between 0.5 and 2 m, and their filling is composed of soft shale, auriferous quartz, antimony sulfide in pockets that reach 1.5 m in width and ferrous pyrite, often in crystallized form [4,6,18].

*2.2. Mining Activity*

The Ribeiro da Serra mine was in operation from 1884 to the beginning of the 20th century, followed by some short intermittent periods of small mining activity. Exploitation was by underground methods.

The gold–antimony deposits were composed of veins that occurred in Schist–Greywacke Complex and derived metamorphic series, where the predominant rocks are shales. A short characterization of these deposits is the following: low-volume deposits, discontinuous, hosted mainly by Variscan granites and Precambrian to Paleozoic metasediments, hydrothermal fluid activity and related in their genesis with granites of the Hercynian age [8].

The average direction of the surrounding rock goes from N to NO, with a slope to the east. The most productive veins occurred in the EO direction, with a pendant to the north (Tapada and Montalto Mines) and NS with a pendant to the west (Ribeiro da Serra Mine).

There were also so-called "thieve veins", usually barren, with a NS direction and pending to the east. They had a particularity, which earned them the name, of affecting the continuity of the main veins, diverting them through failures when they crossed with them.

The thickness of the veins was very variable, varying from a few centimeters to 1.5 m. The presence of antimonite was also irregular, from sparse impregnation to compact media according to well-defined ore shoots. The average content in the mines varied between 5% and 10% of metallic Sb, with average thicknesses of 0.8 m.

The filling of the veins also included auriferous quartz, with an average content of 7 g Au/t, but occasionally also reached very high values, even exposing visible gold [6].

## 3. Materials and Methods

*3.1. Field Sampling*

Two campaigns of soil sampling were performed. The first, performed in 2019, involved an extensive area where 157 samples were collected (Figure 3) in an approximate grid of 35 m by 35 m. It aimed to quantify the chemical composition of existing soils in a comprehensive area that was naturally intensely mineralized and which could be considered as a background for the region. The second sampling campaign (Figure 3), ran in 2022, focused exclusively on the mining heritage area and covered two sub-areas: one to the north where there was probably a stowing of mining wastes (possibly waste rocks), and the other to the south oriented on the right flank of the valley, where almost all mining infrastructure was located. There is a lateral east fork in the south sampling zone to an area where it is assumed there was probably a small tailings disposal. This second sampling was not regular, and the selection of sampling points followed practical criteria applied in situ: preferred zone of waste transport by temporary streams, absence of very dense vegetation, presence of potential mining wastes and ease of excavation for sampling.

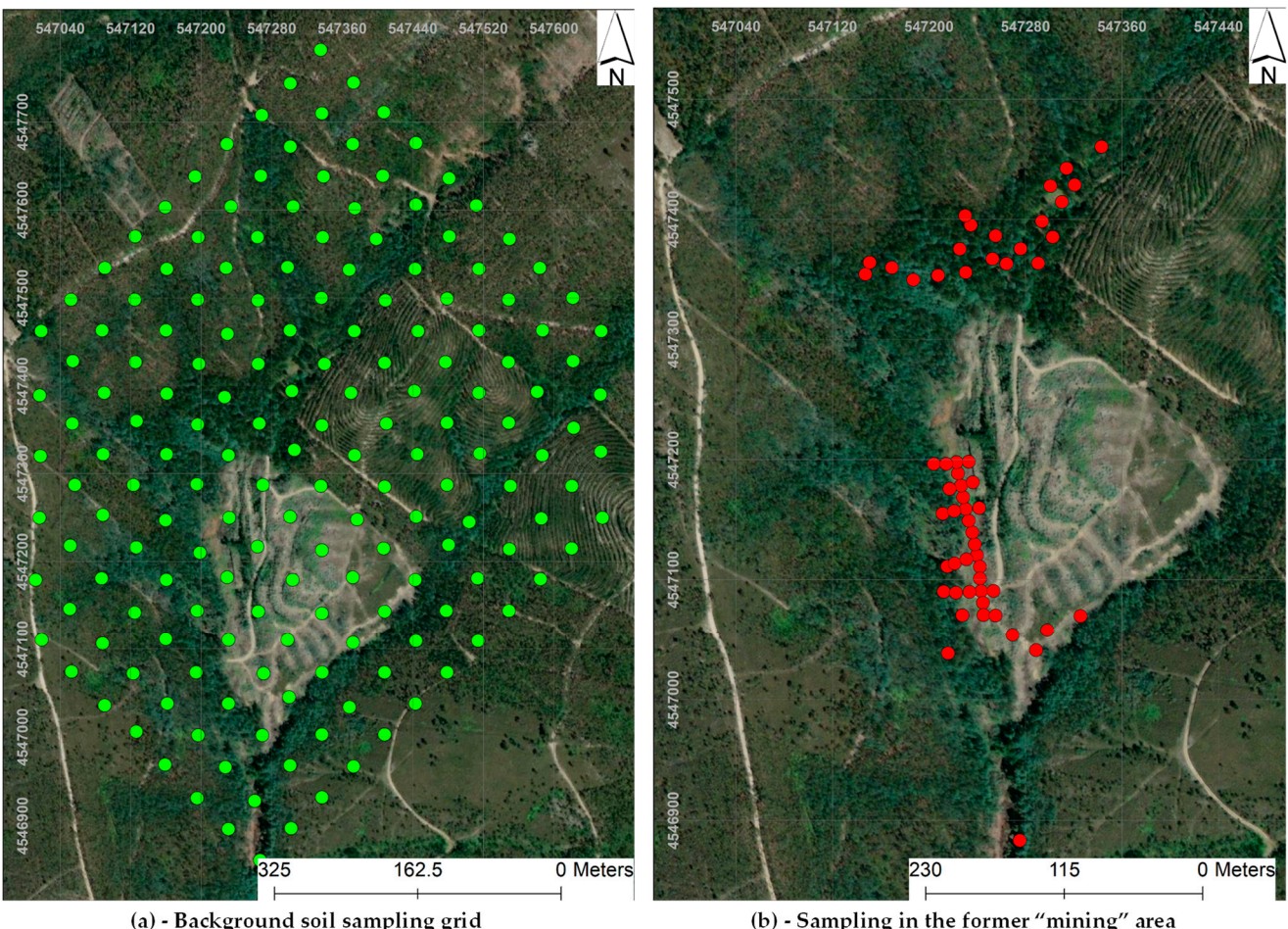

(a) - Background soil sampling grid          (b) - Sampling in the former "mining" area

**Figure 3.** Localization of sampling.

The two background aerial photos (from *World Imagery*) show the intense anthropic action in recent years, mainly by forestry actions.

All soil samples, obtained by pickaxe digging, weighed around 1 kg each and were collected at a depth of 20–30 cm. The soil surface was cleared of superficial debris and vegetation and placed in polyethene bags. Each sample was divided into two halves (around 500 g each) using a sample divider or riffler (Jones splitter). One of the samples was stored. In the other sample, a fraction less than 75 micra was stored in a temperature-controlled environment protected from sunlight and was used for direct chemical analysis when the quantity was sufficient. This fraction of particles less than 75 μm varied between 0% and 30% (around 150 g). The coarse grain size particles (<75 μm) were sub-sampled either by Jones splitter or by manual quartering according to the quantity. When there was not enough quantity of fine particles they were disaggregated in a porcelain mortar, sieved (<2 mm) and homogenized.

Subsequently, they were stored in a temperature-controlled environment protected from sunlight in accordance with the procedures in the manual "Guidance on Sampling and Analytical Methods for Use at Contaminated Sites in Ontario" [19].

### 3.2. Chemical Composition and Environmental Characterization

For the physical characterization, the following parameters were considered and measured: grain size distribution obtained by laser diffraction, moisture, density, porosity and terminal falling velocity in water.

The elemental chemical composition of the soil was obtained by X-ray fluorescence analysis (XRF) using Oxford Instruments X-MET7500 analyser . Each sample was analyzed in duplicate: the first with the soil as collected and the second with the fine fraction of the soil (<0.075 mm) obtained by sieving. Another chemical parameter assessed was the soil pH [20].

For the environmental characterization, the following parameters were assessed: organic carbon, biochemical oxygen demand, natural leaching and acid generation tests. A small fraction of the samples evidenced acid-generation capacity. In the natural leaching tests, the following parameters were analyzed in the leachates: pH, salinity, total dissolved solids, electric conductivity and dissolved oxygen [20].

Grain size distribution was performed in 13 of the 58 samples collected. The coarsest particles in the samples rarely exceeded 15 mm. The grain size distribution evidenced heterogeneity among the samples. The $D_{50}$ of those samples varied between 0.337 mm (minimum) and 12.4 mm (maximum).

The volumetric mass densities mass was determined in 13 samples according to Portuguese Standard NP 83-1965 [21]. The values varied between 2.65 and 2.90.

The soil pH was assessed in 13 samples by two methods—water and calcium chloride. The difference in values obtained by the two procedures was negligible, although it was found that the pH in the solution with $CaCl_2$ was slightly lower than that in the solution with distilled water. The studied soils had acid pH with values between 3 and 5, except for two samples with a pH of around 6.

The organic carbon was assessed in 12 samples by the Walkley-Black Direct Method for soils and sediments. Only three showed concentrations above 2.5%, which is considered the average for soils in the area. The other nine samples had values below 2%, evidencing their probable mining origin.

The biological oxygen demand in soils, measured in 13 samples, exhibited low values between 50 and 70 mg/l for 11 samples and lower values (23 and 35 mg/L) in two samples.

Natural leaching tests, performed in 13 samples with a duration of 72 h, allowed us to conclude that the pH of the water decreased in all samples to values between 3.5 and 6, except for a single sample, whose pH value increased to 7.74, and that the only element that was significantly leached in all the samples was copper.

The Net Acid Generation (NAG) test performed in 13 samples showed that four samples had the potential for acid generation, two were uncertain and seven did not have that potential.

## 4. Results

### 4.1. Preliminary Assessment

For background characterization, only Sb, As, Ti, Pb, Zn and Mn were analyzed in the 157 samples collected for geological survey purposes. In the second survey, designed for environmental intents, the following elements were analyzed in 58 samples collected in the former mining area: As, Ca, Cu, Fe, K, Mn, Mo, Nb, Rb, Sb, Sn, Sr, Ti, Y, Zn and Zr.

Portuguese legislation establishes reference values for various soil types: (a) soils in environmentally sensitive sites, (b) soils less than 30 m from surface waters, (c) shallow soils, (d) stratified soil remediation and (e) unstratified soil remediation. Within each of these categories, the legislation distinguishes between soil with or without groundwater use and within each of these subclasses distinguishes agricultural, urban and industrial uses. The reference values used in this study are for shallow soils without the use of groundwater and for industrial purposes, corresponding to the most permissive values.

Considering this framing of the Portuguese Soil Act, only the following elements are considered soil contaminants: As, Cu, Mo, Sb and Zn. The legislation establishes the following thresholds for these elements, in ppm: As—18; Cu—230; Mo—40; Sb—40; and Zn—340. All the samples showed concentrations below the threshold for Cu, Mo and Zn, while all the concentrations in As and Sb were higher than the legal threshold.

The elements analyzed in common in both surveys were As, Sb and Zn. The values for these elements are displayed in Table 1.

**Table 1.** Comparison of elements of concern in two different areas (concentrations in ppm).

| | Arsenic | | Antimony | | Zinc | |
|---|---|---|---|---|---|---|
| | Background | Mining Area | Background | Mining Area | Background | Mining Area |
| Average | 1501 | 767 | 1077 | 5862 | 32 | 46 |
| Standard Deviation | 1556.1 | 848.9 | 3661.2 | 7566.0 | 51.2 | 37.0 |
| Variance | 2,421,588 | 720,577 | 13,404,603 | 57,243,607 | 2623 | 1368 |
| Maximum | 6775.0 | 3430.0 | 31247.3 | 27204.0 | 443.7 | 185.0 |
| Minimum | 8.7 | 58.0 | 0.0 | 83.0 | 6.3 | 5.0 |
| Median | 1263.3 | 409.0 | 127.3 | 2289.0 | 15.3 | 34.0 |
| Logarithm Average | 6.95 | 6.24 | 5.72 | 8.19 | 2.83 | 3.58 |
| Logarithm Variance | 0.73 | 0.80 | 2.53 | 0.98 | 1.27 | 0.50 |

*4.2. Comparison of Populations*

Background and mined area sample populations were compared considering their means and correlations matrices. The comparison of means (As, Sb, Zn) was performed by the Hotelling test for two populations with sizes $N_1$ and $N_2$, with mean vectors $\bar{X}_1$ and $\bar{X}_2$ and variance-covariance matrices $S_1$ and $S_2$ at a defined level of significance (0.01 and 0.05). In this case, the Hotelling $T^2$ test takes the form

$$T^2 = \frac{N_1 N_2}{N_1 + N_2} . (\bar{X}_1 - \bar{X}_2)' . S^{-1} . (\bar{X}_1 - \bar{X}_2) \tag{1}$$

where S is the variance-covariance matrix of the pooled population

$$S = \frac{1}{N_1 + N_2 - 2} . [(N_1 - 1).S_1 + (N_2 - 1).S_2)] \tag{2}$$

The meaning of the Hotelling test can be evaluated by the transformation

$$F = \frac{N_1 + N_2 - n - 1}{(N_1 + N_2 - 2)n} . T^2 \tag{3}$$

This methodology applied to the samples of both populations and evidenced that the populations have different means at both levels of significance.

The next step was to verify the homoscedasticity of the distributions, or the equality of the variance-covariance matrices, through the "test of generalized variances", using the so-called homogeneity test of Bartlett.

In this test, the hypothesis $H_0$: $\Sigma_1 = \Sigma_2 = ... = \Sigma_k$ of equality of the variance-covariance matrices of k normal populations was tested against the opposite hypothesis that postulated its difference. Let $S_i$ be an unbiased estimate of $\Sigma_i$ with $N_i$ degrees of freedom, where $N_i = n_i - 1$ and $n_i$ the number of observations (samples) in each vector (sub-population). When $H_0$ $_0$ is true

$$S = \frac{1}{(\sum_{i=1}^{k} n_i) - k} . \sum_{i=1}^{k} (n_i - 1) S_i \tag{4}$$

is the pooled estimate of the common variance-covariance matrix. The statistical test is

$$M = \left[ \left( \sum_{i=1}^{k} n_i \right) - k \right] . \ln |S| - \sum_{i=1}^{k} [(n_i - 1). \ln |S_i|] \tag{5}$$

and BOX (1949) demonstrates that if the scale factor

$$C^{-1} = 1 - \frac{2m^2 + 3m - 1}{6(m+1)(k-1)} \cdot \left( \sum_{i=1}^{k} \frac{1}{n_i - 1} - \frac{1}{\sum_{i=1}^{k} n_i - k} \right) \tag{6}$$

is considered, the quantity $\mathbf{MC^{-1}}$ approximately follows a chi-square distribution with degrees of freedom equal to 1/2. (k−1). m.(m+1). In previous expressions |S| represented the determinant of matrix S. The test is therefore based on the difference between the logarithm of the determinant of the covariance matrix of the pooled populations and the mean of the logarithms of the determinants of the variance-covariance matrices of each subpopulation.

The test was applied at a 0.05 level of significance to both populations, the background and the mining heritage areas. The conclusion was the heteroscedasticity of the populations, proving that the mining area is relatively distinct from the background.

*4.3. Characterization of the Legacy Area*

The distributions of As and Sb are both log-normal and hypothesis verified by the Kolmogorov–Smirnov test at 99% confidence level. Figure 4 represents the histograms of the logarithm of the concentrations for As and Sb and the graphical results of the Kolmogorov–Smirnov test.

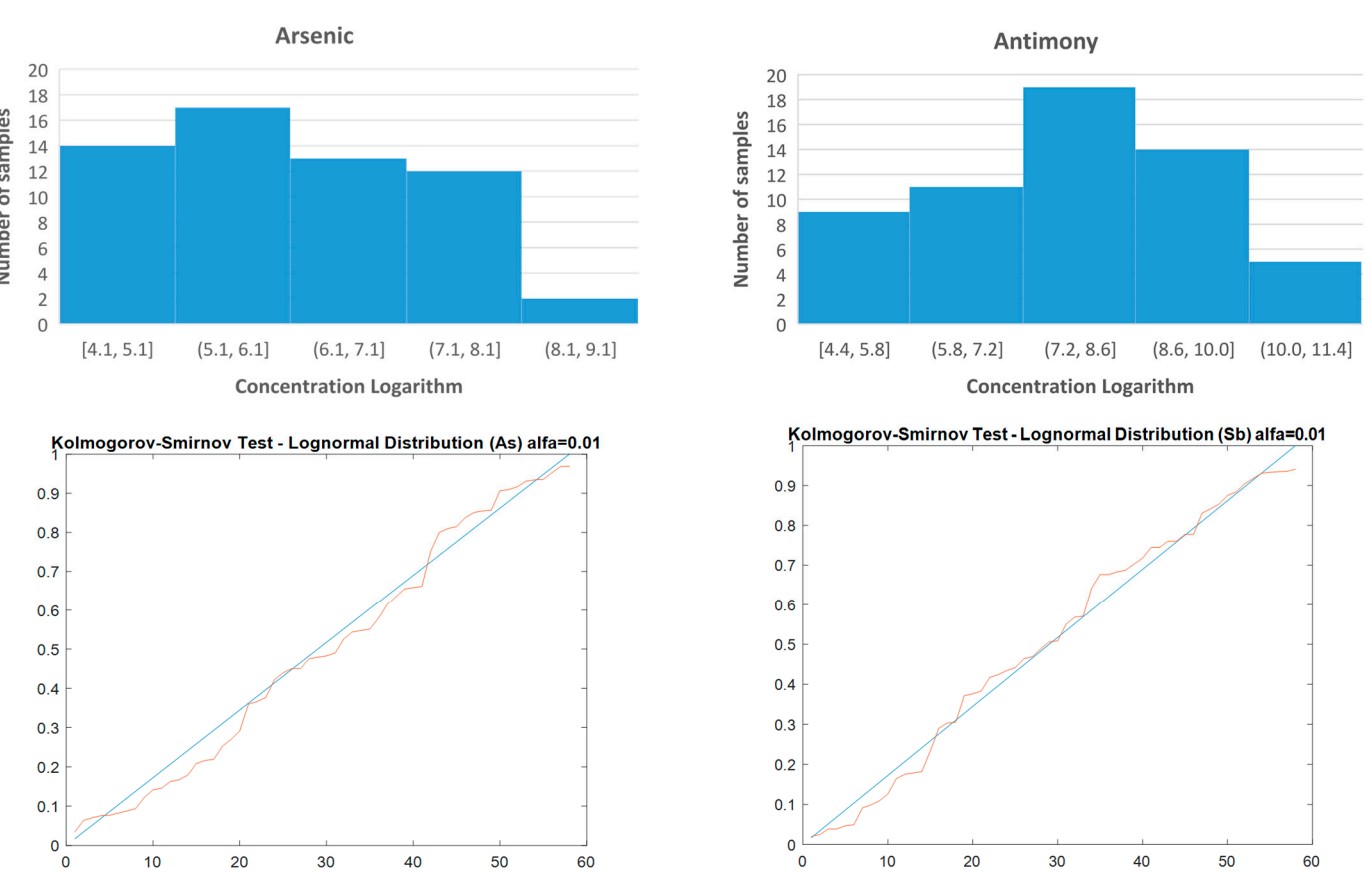

**Figure 4.** Histograms of the logarithm of the concentrations (As and Sb in ppm) and Kolmogorov–Smirnov Tests.

The lognormality of the distributions justifies the former application the Bartlett test, which is very sensitive to non-normality.

The Pearson and the Spearman correlation coefficients between the principal contaminant elements are shown in Table 2, Pearson in the upper triangle, and Spearman [22]

in the lower. Tin was also included in these correlation matrices due to the high correlation that was evidenced with antimony and partially with arsenic.

The critical Pearson correlation coefficient at a 99% confidence level for six variables and 50 degrees of freedom was 0.479, while the equivalent critical Spearman correlation coefficient, also at a 99% confidence level, was 0.4960. So, we can consider as significant the following correlation coefficients: Pearson: Sb-Sn, As-Sb; Spearman: the same as before as well as the association As-Sn.

**Table 2.** Correlations (Pearson and Spearman) between elements of potential environmental concern. The significant ones at a 95% confidence lever are in bold.

|       | **As**  | **Cu**  | **Mo**  | **Sb**  | **Sn**  | **Zn**  |
|-------|---------|---------|---------|---------|---------|---------|
| As    |         | −0.2559 | −0.2    | **0.5303** | 0.4636 | −0.2641 |
| Cu    | 0.2345  |         | 0.1235  | −0.0329 | 0.0461  | 0.0742  |
| Mo    | −0.1294 | 0.2077  |         | −0.2892 | −0.2668 | 0.1953  |
| Sb    | **0.8303** | −0.0246 | −0.1931 |         | **0.9569** | −0.2772 |
| Sn    | **0.7964** | 0.2046  | 0.0780  | **0.9318** |         | −0.2752 |
| Zn    | −0.060  | 0.5514  | 0.1231  | −0.1231 | −0.1954 |         |

The histogram of arsenic concentrations (Figure 5) clearly shows that low concentrations of As occur in the waste rock area (orange color).

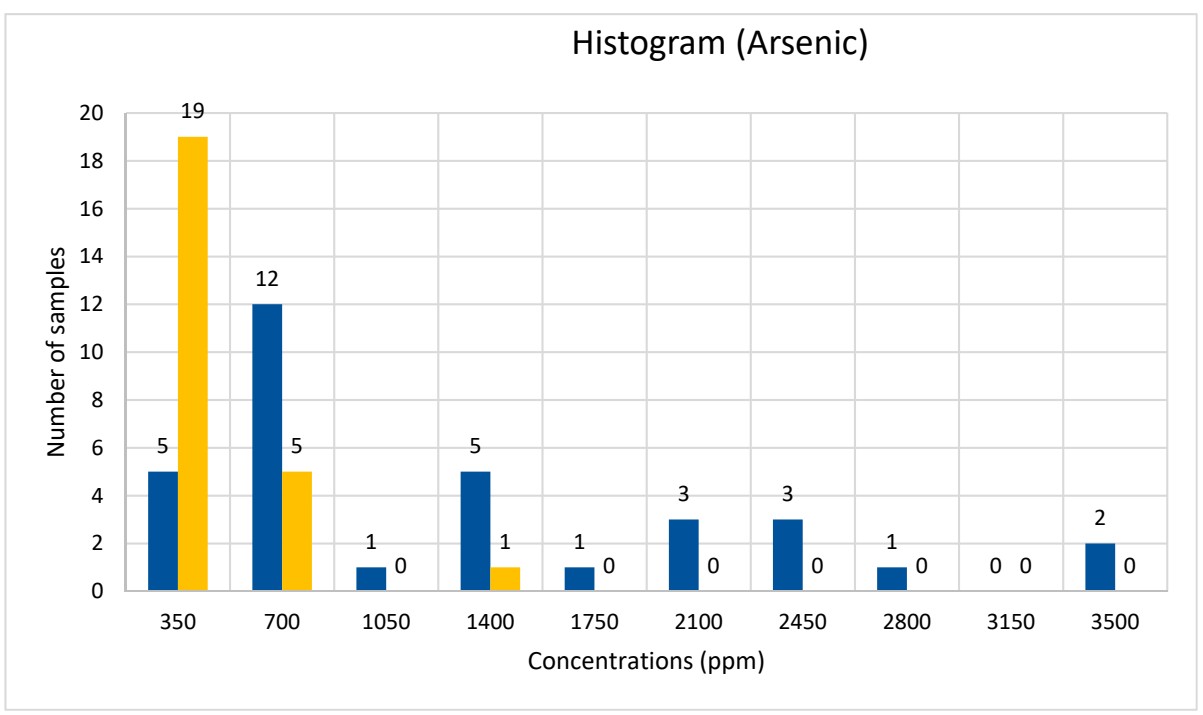

**Figure 5.** Histogram of arsenic concentrations (mining area in blue and waste rock in orange).

A similar distribution occurs for antimony displays in Figure 6.

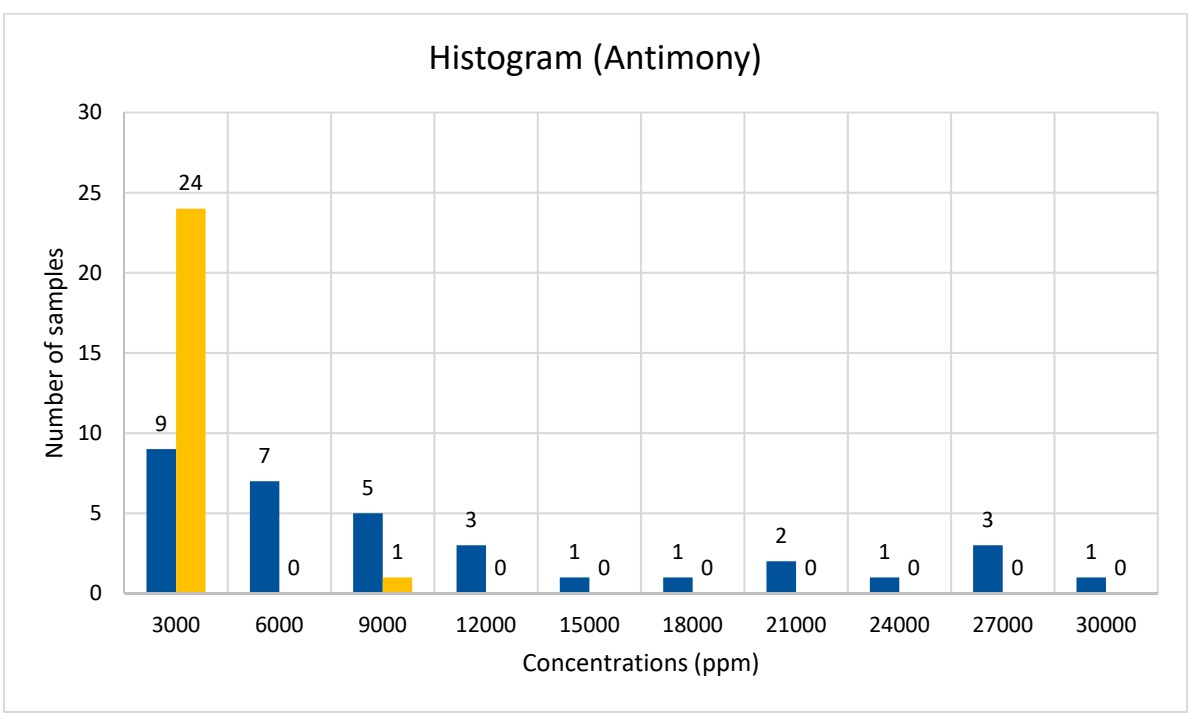

**Figure 6.** Histogram of antimony concentrations (mining area in blue and waste rock in orange).

### 4.4. Space Distribution of Concentrations

The spatial distribution of concentrations was determined by ordinary kriging using a semi-variogram with a spherical nested structure, using the software GMS (*Groundwater Modeling System*). The contributions and range were assessed for each element, as well as the nugget effect that had minor importance in all the studied cases. Figure 7 shows the spatial distribution of antimony and arsenic in the study area. It is noticeable that there was a great similarity in the spatial distribution of both elements: the waste rock area had minor concentrations when compared with the concentrations in the valley, where the mining infrastructures and the tailings disposal existed.

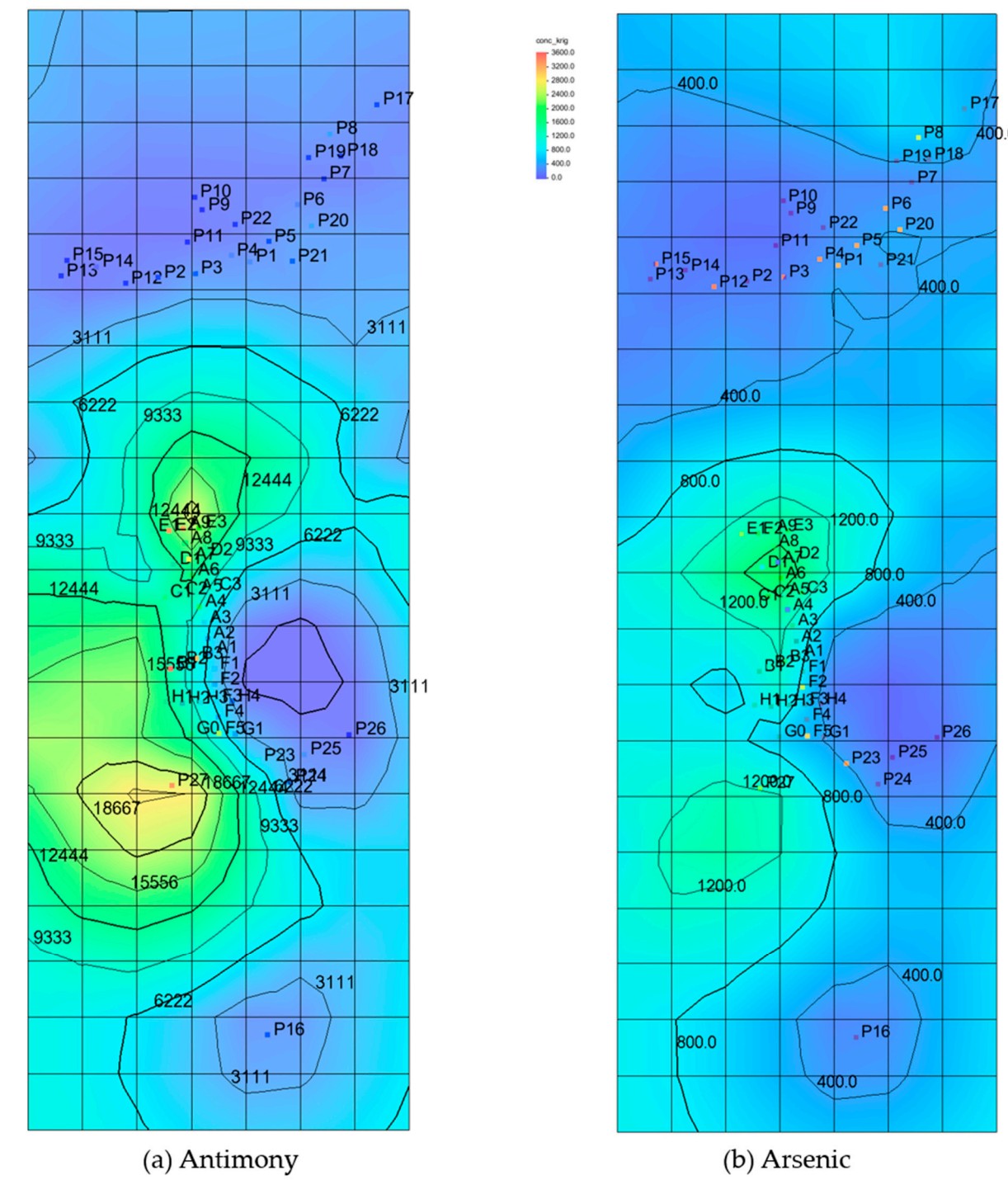

**Figure 7.** Space distributions of antimony and arsenic (ppm) by ordinary kriging.

*4.5. Principal Component Analysis*

Several similar multivariate statistical approaches are available for cases when the relationship between the variables has no causality pattern known in advance. These methods are generally designated as factor analysis as they attempt to determine the hidden factors that generate the dependence or variation of responses. Conceptually, there is the need to analyze phenomena characterized by many variables that are correlated to a greater or lesser degree. These correlations are like a veil that prevents the proper evaluation of the role of each variable in the studied phenomenon. The principal component analysis (PCA) allows moving to a new set of variables the main components that have

the advantage of not being correlated with each other and that can be ordered considering the information they have incorporated [23–26].

Their variance is used to measure the amount of information incorporated into a component. The higher the variance, the greater the amount of information stored in the main component. For this reason, the main components were selected in descending order of variance. Generally, the main components are extracted from normalized variables to avoid problems derived from different scales. If p variables are standardized, the sum of variances is equal to p since the variance of each standardized variable is unitary.

The new set of variables obtained by the principal components method is equal to the original number of variables, and their variance is equal to the sum of the original variances. The big difference lies in the fact that the main components are calculated uncorrelated with each other, allowing most of the variability to be explained with few components. The main components are expressed as linear combinations of the original variables. The algorithm used was written in Matlab (version R2020a, The Mathworks Inc., USA) (version, manufacturer, city, state abbreviation if USA or Canada, country).

Concerning the three main axes, the overall contribution of the first axe was 0.4393, while the contributions with two or with three axes were, respectively, 0.6281 and 0.7736. The samples collected in the mining area are represented by red circles, while the samples collected around the waste rock disposal are represented by blue x-marks (Figure 8).

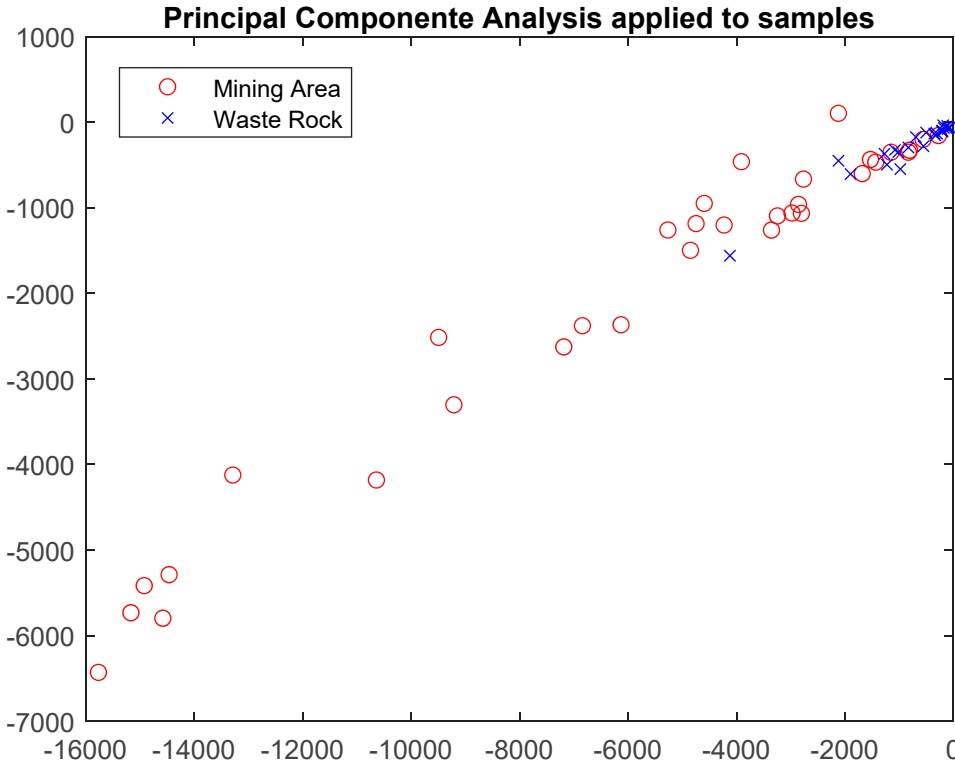

**Figure 8.** PCA—Projection of samples in the plane defined by the two main principal components.

The principal component analysis was also applied to the variables through a loading plot (Figure 9) using the software AnDad [27].

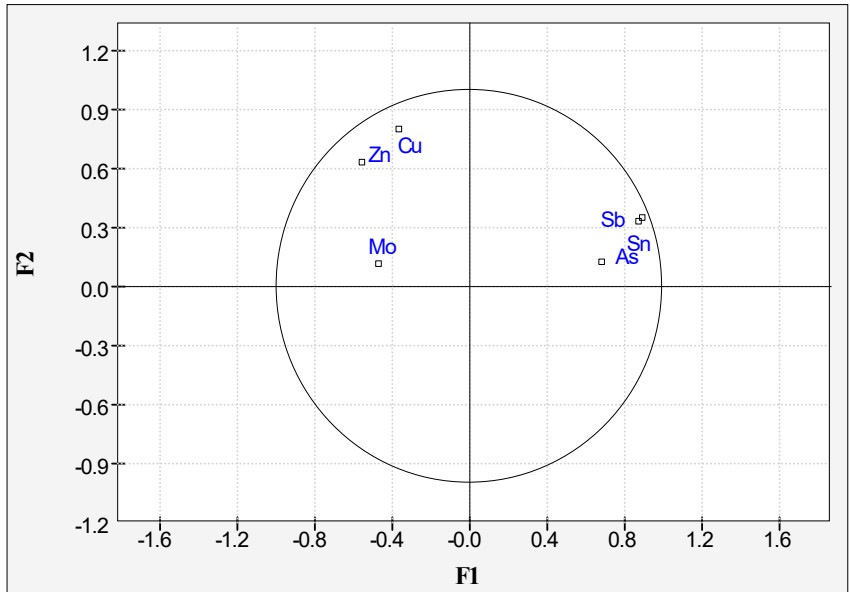

**Figure 9.** Principal components analysis, projection of variables in the first factorial plan F1, F2 (software ANDAD, vs 7.12, CVRM/IST 2000, Lisbon, Portugal).

The association between Sb-As-Sn becomes evident as well as a large positive association of these elements with the first factorial component. The second component has a large positive association with Zn and Cu.

*4.6. Correspondence Analysis*

Correspondence analysis allows performing simultaneous R-mode and Q-mode factor analysis. The theory was extensively developed by Benzécri [28] and later by Legendre [29] and Greenacre [30].

The correspondence analysis proceeds by operating in an inferred matrix from a table of contingencies that is transformed so that its elements can be considered conditional probabilities. Because of the nature of the transformation, the relationships between rows and columns of the transformed table are the same as those that existed in the original data array.

Since, in general, the grand total (as well as the row totals) will consist of a mixture of various types of measurements, the transformation process is strongly dependent upon units of measurement. Despite this objection, correspondence analysis is routinely applied to interval and ratio data sets. In these applications, the transformation process is envisaged as no more than an arbitrary procedure designed to close the data set and to ensure that rows and columns of the data matrix are scaled in equivalent manners when computing either R-mode or Q-mode factor analysis. When used in this manner, the similarity measurement is often referred to as the "profile distance"; it reflects the relative magnitude of the variables rather than their absolute values [31]. This similarity measurement is expressed by

$$r_{jk} = \sum_{i=1}^{n} \left( \frac{P_{ij} - P_i.P_j}{\sqrt{P_i.P_j}} \right) \left( \frac{P_{ik} - P_i.P_k}{\sqrt{P_i.P_k}} \right) \tag{7}$$

$P_{ij}$ is the observed probability in row i and column j of the body of the contingency table, and $P_i$ and $P_j$ are the expected probability computed as the product of marginal probabilities.

The initial data table [X] contains n rows representing soil samples and m columns of variables (concentrations in chosen elements). The elements of [X] are converted into

joint probabilities by dividing each element of the matrix by the total, which is the scalar $\Sigma\Sigma x_{ij}$, originating a new matrix [B]

$$[B] = \frac{1}{\sum\sum x_{ij}}[X] \tag{8}$$

Then, a square matrix [M] (mxm) is created that contains the totals of the columns of [B] on the main diagonal and whose remaining elements are null. Likewise, we define a square matrix [N] (nxn) that contains the sum of the elements of the lines of [B] on the main diagonal, the remaining elements being null. These two matrices contain the marginal probabilities of the columns and rows and can be used to transform [B]

$$[W] = [N]^{-1/2}.[B].[M]^{-1/2} \tag{9}$$

Since these are diagonal matrices, the operations $[N]^{-1/2}$ and $[M]^{-1/2}$ are equivalent to replacing each element in the main diagonal by $1/\sqrt{n_{ij}}$ and $1/\sqrt{m_{ij}}$. Matrix [W] will be (nxm), with a transformed $w_{ij}$ element corresponding to each original $x_{ij}$ element. The array of the crossed products of the columns is simply (the symbol T denotes transpose)

$$[R] = (W)^T.[W] \tag{10}$$

Likewise, the matrix of the cross-products of the lines is

$$[Q] = [W].[W]^T \tag{11}$$

The eigenvalues of [R] and [Q] must be identical, except that Q will have *n-m* additional eigenvalues, but with null values.

The eigenvectors of [R] can be converted into loadings of the correspondence factors by multiplying each vector by the corresponding singular value which is the square root of the corresponding eigenvalue. That is

$$R - mode\ loadings = \sqrt{\lambda}\ .mode - R\ eigenvector \tag{12}$$

In matrix notation, the singular values of [R] can be interpreted as occurring diagonally on a diagonal square matrix [$\Lambda$] (mxm). The eigenvectors of [R] form the columns of a [U] matrix (mxm). The matrix equation used to determine the loadings in R-mode is then

$$[A^R] = [U].[\Lambda] \tag{13}$$

There is still the need to reduce to a common metric, which is conducted by the matrix product

$$[\bar{A}^R] = [M]^{0.5}.[A^R] \tag{14}$$

The scores of each of the n observations in the m correspondence factors are simply

$$[S^R] = [W].[\bar{A}^R] \tag{15}$$

These scores can be represented along the axes defined by the correspondence factors in R-mode, in the same way as the factor loadings in the main components.

The matrix equation used to determine the loads in Q-mode is then

$$[A^Q] = [V].[\Lambda] \tag{16}$$

It is reduced to the same metric by changing the scale

$$[A^Q] = [N]^{0.5}.[A^Q] \tag{17}$$

The scores of each of the n observations in the m correspondence factors is simply

$$[S^Q] = [W]^T.[\bar{A}^Q] \tag{18}$$

This algorithm for correspondence analysis was implemented in Matlab and then applied to the data collected in both areas.

The plots of the objects (samples) and of the variables in the same metric is represented in Figure 10.

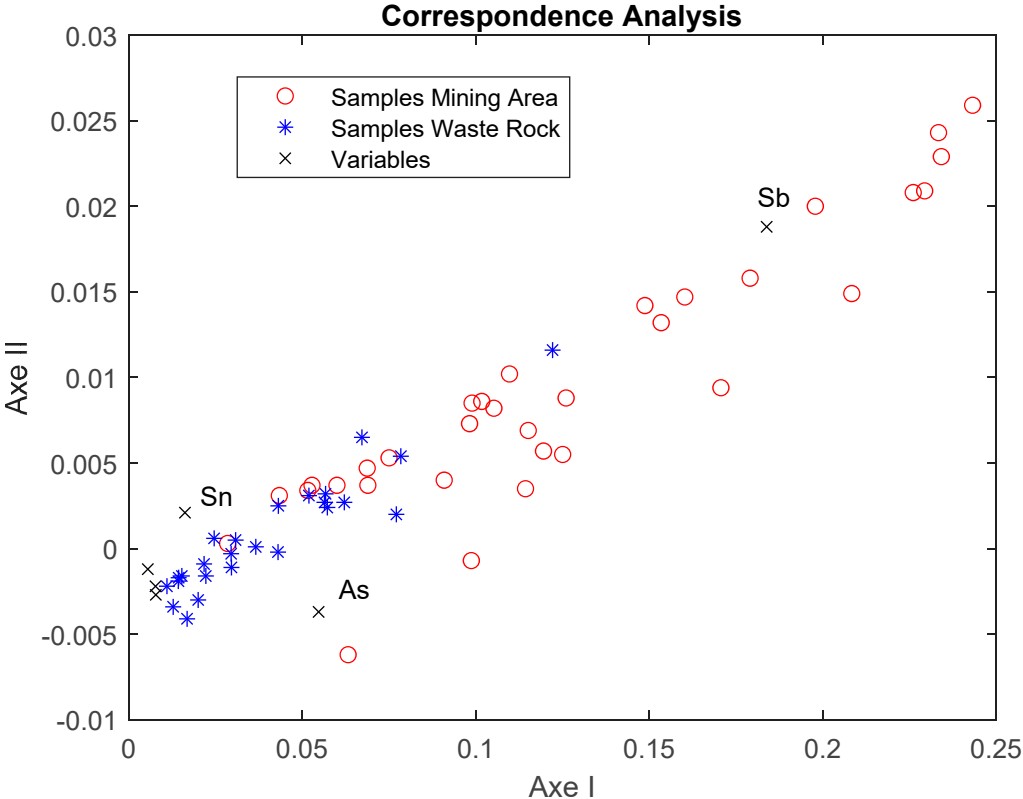

**Figure 10.** Plots of samples and chemical elements in the same metric using correspondence analysis.

The distinction between the two areas of sample collection is well-defined. The former mining area is characterized by high concentrations of antimony, while the waste rock zone is individualized by the concentrations of arsenic and tin, this one to a lower extent. As previously demonstrated, the concentrations of arsenic in the mining area are lower than in the background, where a regional anomaly exists.

## 5. Toxicity of Antimony and Arsenic

Reliable data concerning the toxicology of antimony and arsenic are available in the Risk Assessment Information System (RAIS) and they are reported in Table 3.

**Table 3.** Toxicological data for antimony and arsenic.

| Type of Dose | Antimony, Metallic | Arsenic, Inorganic |
|---|---|---|
| Inhalation Acute Reference Concentration (mg/m³) | 0.001 | 0.0002 |
| Inhalation Chronic Reference Concentration (mg/m³) | 0.0003 | 0.000015 |
| Inhalation Subchronic Reference Concentration (mg/m³) | 0.001 | - |
| Inhalation Short-term Reference Concentration (mg/m³) | 0.001 | - |
| Oral Acute Reference Dose (mg/kg-day) | 1 | 0.005 |
| Oral Chronic Reference Dose (mg/kg-day) | 0.0004 | 0.0003 |
| Oral Subchronic Reference Dose (mg/kg-day) | 0.0004 | 0.005 |
| Oral Slope Factor (mg/kg-day)⁻¹ | - | 1.5 |

Antimony is toxic by inhalation and ingestion for all durations of exposure, from acute to chronic. It is not carcinogenic. Arsenic is also toxic by inhalation and ingestion,

but it is carcinogenic when ingested. Inhalation of particles susceptible to reaching the pulmonary alveolus is very unlikely because the fraction of particles with grain size less than 10 μm was either null or very small according to the measurements of the grain size distribution made on the soil samples collected. The maximum fraction of PM10 found was around 0.03.

Contamination by ingestion is also very unlikely once the area does not produce edible crops.

## 6. Discussion

The environmental legacy of this former mine, typical of those exploited in the area in the 19th and early 20th centuries, is constituted by the alterations in the geomorphology of the site and in the composition of the environmental compartments due to open-air storage of mine wastes, tailings and roasting wastes. In this location, and in other antimony mines in the same area, there are no visible signs of these environmental impacts, as is generally the case in most abandoned mines:

- there is no acid mine drainage;
- waste storage sites are indistinguishable from the landscape and their location was only detected after research with advanced technologies used in this project (LIDAR survey and digitalization of old maps);
- the ruins of industrial activity are minimal (ruins of a stack and two administrative buildings) and could have had another origin (Figures S2 and S3).

Nothing apparently indicates there were mines in this location, covered by intense vegetation. One of the objectives of the project was to verify whether there really would exist a delimited environmental legacy, although invisible to the eye, but distinguishable through soil analysis.

Physical, chemical and environmental characterization of the samples collected allowed us to conclude that the soil is acidic, not suitable for agriculture due to its low organic matter content, evidencing its mining origin, and also that natural leaching produces slightly acidic solutions and that an acid generation potential only exists in a minority of samples.

It was found that out of the five elements existing in the soil with potential environmental impact as defined in the Portuguese Legislation, only two (As and Sb) always exceeded the thresholds, while all others (Cu, Mo, Sn and Zn) always occurred at lower concentrations. The distribution of the concentration of the main contaminants was lognormal. The environmental legacy was recognizable in relation to the background concentrations for their difference in chemical composition, especially regarding the pernicious elements noticed. The legacy area has higher concentrations of Sb and lower of As in relation to the background. These two elements are correlated in the legacy area but not in the background. The average value of Sb grades was very high—5862 ppm—when compared with normal concentrations in soils 0.3–8.4 mg kg$^{-1}$ [11] or to the values found in other former mines in Portugal: Sarzedas mine (663.1 mg/kg) [15]) and in the São Domingos copper sulfide mine (467.8 mg/kg) [16]. It was also very high when compared with wastes from other former antimony mines in the world, being only comparable to the concentrations found in Tuscany (Italy) (5598 mg/kg, Baroni [32]). The average As concentration (767 ppm), although lower than the sampled background (1501 ppm), was much above the toxicity values established worldwide for arsenic.

In the area where the environmental legacy was appraised, two complementary distinct zones were distinguished by multivariate statistical methods: the "mining" area, with higher concentrations in Sb and As, and the area where there was possibly storage of waste rock, with much lower concentrations in these two elements.

The treatment of the information (chemical analysis of As, Cu, Mo, Sb, Sn and Zn in soil samples) by principal component analysis and correspondence analysis allowed the confirmation of the existence of these two distinct areas.

The former mining area analyzed in this work is anomalous in relation to the Valongo Mineral Belt, which in turn is a natural anomaly in relation to a more comprehensive geological environment such as those used to define the background levels on which environmental legislations are based. It will be interesting to study, as a continuation of this work, how quantitative global contamination indices of potentially toxic elements and mining wastes vary (there is an exhaustive and critical listing of these indices in [33]) when the context in which the background is defined is altered.

## 7. Conclusions

Based on all the research developed the following conclusions may be stressed:

- The mining heritage area is discernible from the highly mineralized background, having higher concentrations of antimony and zinc and lower concentrations of arsenic. This fact was statistically demonstrated by statistical tests: equality of vector means (Hotelling) and equality of variance-covariance matrices (Bartlett).
- The mining heritage zone still reveals the area where the extraction, the ore processing and the roasting activities took place. There is a net separation from the mining waste disposal zone, discernible by higher concentrations of both antimony and arsenic.
- Natural dispersal processes, especially the aqueous transport of particles by runoff through streams, contributed to an environmental dispersion of particles with antimonite, but to a minor extent and in a relatively small area. The reason lies in the fact that the solid wastes have a small exposure to the atmosphere, being most of the tailing buried, although superficially, and especially because the topography did not create valleys fed by permanent and intense rainwater drainage basins.
- The environmental legacy was incorporated into the new post-industrial landscape due to the intense planting of eucalyptus trees that absorb the infiltrated water and strongly decrease the intensity of runoff. This forest of anthropic origin also contributed to the fixation of the soil particles avoiding intense dispersion. This assertion becomes evident when one examines the spatial distribution of Sb concentrations obtained by the survey of the background of the area, which shows that the anomaly is restricted to the area where there was mining activity (Figure S1) (Supplementary Material #1). The physical dispersion of mining waste is restricted to 300 m.

**Supplementary Materials:** The following supporting information can be downloaded at: https://www.mdpi.com/article/10.3390/min13020257/s1, Figure S1: Global distribution of Sb; Figure S2: Location of ruins at the site; Figure S3: Picture of administrative building and chimney at the end of the 19th Century; Table S1: Concentrations in the mining legacy area.

**Author Contributions:** Conceptualization: A.F. (António Fiúza) and A.F. (Aurora Futuro); methodology: A.F. (António Fiúza) and A.F. (Aurora Futuro); statistical analysis: A.F. (António Fiúza) and J.G.; software: A.F. (António Fiúza) and J.G.; on-site field work: S.C., A.F. (Aurora Futuro), M.L.D. and A.F. (António Fernandes); geology: A.F. (Aurora Futuro); laboratory testing: M.L.D., C.V. and A.F. (António Fernandes); writing—review and editing: A.F. (António Fiúza), A.F. (Aurora Futuro) and S.C.; project administration: A.F. (António Fiúza) and A.F. (Aurora Futuro); funding acquisition: A.F. (António Fiúza). All authors have read and agreed to the published version of the manuscript.

**Funding:** This paper is a result of the project "SHS—Soil health surrounding former mining areas: characterization, risk analysis, and intervention", with the reference NORTE-01-0145-FEDER-000056, supported by Norte Portugal Regional Operational Programme (NORTE 2020), under the PORTUGAL 2020 Partnership Agreement, through the European Regional Development Fund (ERDF).

**Data Availability Statement:**

**Acknowledgements:** The authors thank Project AUREOLE, ERA/MIN/0005/2018, funded by the Foundation for Science and Technology, and framed within the scope of the activities, of GI 3 of the

ICT (UIDB/04683/2020) for the permission to use the data obtained in the soil survey used to define the regional background.

**Conflicts of Interest:** The authors declare no conflicts of interest.

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
