# Peer review of "The Legacy of Potential Environmental Soil Contamination in an Antimony Mining Heritage Area"

_minerals, doi:10.3390/min13020257_

Round 1

Reviewer 1 Report

The article is devoted to the actual problem of assessing the current impact of ancient mining. Dividing of the geochemical background of the deposit area  and the impact of mining is a complicated methodological task. The article is mainly devoted to the statistical description of the data on the content of As, Sb and Zn in soils of background, mining area and waste rock.

There are some questions. According to the lines 159-163 It was analyzed soil parameters, but in the text the soil description and analyses of soil parameters are absent. This information may be very important for explanation of metals content and metal distribution. Also this information could be interesting for other scientists who research soils of similar deposits. The last item of conclusions (390-393 lines) doesn't confirm by the text and data.

One remark - no link in the text to the figure 9. 

The manuscript could be accepted after major revisions.

Author Response

  • According to the lines 159-163 It was analyzed soil parameters, but in the text the soil description and analyses of soil parameters are absent. This information may be very important for explanation of metals content and metal distribution. Also this information could be interesting for other scientists who research soils of similar deposits.

Answer: The following sentences were added to the text (3.2) according to your suggestion:

“Grain size distribution was performed in 13 of the 58 samples collected. The coarsest particles in the samples rarely exceed 15 mm. The grain size distribution evidenced heterogeneity among the samples. The D50 of those samples varied between 0.337 mm (minimum) and 12.4 mm (maximum).

The volumetric mass densities mass was determined in 13 samples according to Portuguese Standard NP 83-1965. The values varied between 2.65 and 2.90.

The soil pH was assessed in 13 samples by two methods – water and calcium chloride. The difference in values obtained by the two procedures is despicable, although it is found that the pH in the solution with CaCl2 is slightly lower than in the solution with distilled water. The studied soils have acid pH with values between 3 and 5, except for two samples with a pH of around 6.

The organic carbon was assessed in 12 samples by the Walkley-Black Direct Method for soils and sediments. Only three showed concentrations above 2.5% which is considered the average for soils in the area. The other nine samples had values below 2% evidencing their probable mining origin.

The Biological Oxygen Demand in soils, measured in 13 samples, exhibited low values between 50 and 70 mg/l for 11 samples and lower values (23 and 35 mg/L) in two samples.

Natural leaching tests, performed in 13 samples with a duration of 72 h, allow us to conclude that the pH of the water decreased in all samples to values between 3.5 and 6, except for a single sample, whose pH value increased to 7.74 and that the only element that was significantly leached in all the samples was copper.

 The Net Acid Generation (NAG) test performed in 13 samples showed that four samples had the potential for acid generation, two were uncertain and seven did not have that potential.

  • The last item of conclusions (390-393 lines) doesn't confirm by the text and data.

Answer – An additional explanation was added to the text:

The environmental legacy was incorporated into the new post-industrial landscape due to the intense planting of eucalyptus trees that absorb the infiltrated water and strongly decrease the intensity of runoff. This forest of anthropic origin also contributed to the fixation of the soil particles avoiding intense dispersion. This assertion becomes evident when one examines the spatial distribution of Sb concentrations obtained by the survey of the background of the area, which shows that the anomaly is restricted to the area where there was mining activity (Supplementary Material),

 Suplementary material #1

  • One remark - no link in the text to the figure 9.

Answer – The link was included.

Reviewer 2 Report

The abstract should be better structured. Abstract is too long

The novelty and scientific quality of manuscript is poor.

Keywords: “Antimony” is not appropriate, it is already in the title.

The introduction section is limited to presenting the description of the study site.

The section for sample description was not sufficient. When were the field campaigns carried out? what was the sampling procedure? In addition, analytical as well as quality control methods were not mentioned clearly in this section. The author needs to mention those above points to confirm accuracy of the data and experiments.

The Results section needs to be enhanced. The discussion is extremely poor, more research in similar studies is needed. More discussions about the results should be carried out based on references.

Line 29. Close parentheses

 The author needs to revise the citation format. For example, Line 43. “([1], [2] and [3])”. Review the author guide and correct citations throughout the manuscript.

Fig 1. Use commas to separate thousands for numbers with five or more digits. The author should improve the quality of the map.

join figures 3 and 4, use literals (a) and (b)

The quality of all images should be improved

Section 4 "Data analysis" should be called "results and discussion"

Line 178 – 206, this corresponds to methodology

Line 232 “A similar distribution occurs for antimony displays in Figure 7” place this text in the previous paragraph.

The chart legend differs in size for figures (a) and (b). Unify the size of the text and improve the quality of the figures.

Section 4.6 "correspondence analysis" (line 283-337) is very theoretical, its content corresponds to methodology

Separate discussion and conclusions

The conclusions should be more concise

Author Response

  • “Antimony” is not appropriate, it is already in the title.

Answer – It was replaced by “Valongo mineral belt”.

  • The introduction section is limited to presenting the description of the study site.

Answer – The introduction was rewritten and amplified.

  • The section for sample description was not sufficient. When were the field campaigns carried out? what was the sampling procedure? In addition, analytical as well as quality control methods were not mentioned clearly in this section. The author needs to mention those above points to confirm accuracy of the data and experiments.

Answer: the dates of the campaigns are now included in the text.

The sampling procedure is included in the new text:

“All soil samples, obtained by pickaxe digging, weighed around 1 kg each and were collected at a depth of 20-30 cm. The soil surface was cleared of superficial debris and vegetation and placed in polyethene bags. Each sample was divided into two halves (around 500g each) using a sample divider or riffler (Jones splitter). One of the samples was stored. In the other sample, the fraction less than 75 micra was stored in a temperature-controlled environment protected from sunlight and was used for direct chemical analysis when the quantity was sufficient. This fraction of particles less than 75 mm varied between 0 and 30% (around 150 g). The coarse grain size particles (<75 mm) were sub-sampled either by Jones splitter or by manual quartering according to the quantity. When there wasn´t enough quantity of fine particles they were disaggregated in a porcelain mortar, sieved (<2 mm) and homogenized.

Subsequently, they were stored in a temperature-controlled environment protected from sunlight in accordance with the procedures in the manual “Guidance on Sampling and Analytical Methods for Use at Contaminated Sites in Ontario” [19].”.

The analytical and quality control methods are mentioned in point “3.2. Chemical composition and environmental characterization”

 The Results section needs to be enhanced. The discussion is extremely poor, more research in similar studies is needed. More discussions about the results should be carried out based on references.

Answer – The Results and the Discussion sections were rewritten.

  • Line 29. Close parentheses

Answer – Done.

  • The author needs to revise the citation format. For example, Line 43. “([1], [2] and [3])”. Review the author guide and correct citations throughout the manuscript.

Answer – The citation format was changed according to the author’s guide.

  • Fig 1. Use commas to separate thousands for numbers with five or more digits. The author should improve the quality of the map.

Answer – The quality of the map was improved.

  • join figures 3 and 4, use literals (a) and (b)

Answer – The suggestion was done.

  • The quality of all images should be improved

Answer: Que quality of almost all the figures was improved.

  • Section 4 "Data analysis" should be called "results and discussion"

Answer: The title of the section was changed

  • Line 178 – 206, this corresponds to methodology

Answer: Lines 178-206 define the nature of the available data and the elements that will be considered contaminants, according to the applicable legislation. The methodology is the definition of methods, i.e. the selection of the paths to reach a certain scientific target. Thus, the referred lines indicate the set of information that will be considered, and why, and not the method that will be used to make this study.

  • Line 232 “A similar distribution occurs for antimony displays in Figure 7” place this text in the previous paragraph.

Answer: The order was changed, as suggested.

  • The chart legend differs in size for figures (a) and (b). Unify the size of the text and improve the quality of the figures.

Answer: The legend was unified and its quality improved.

  • Section 4.6 "correspondence analysis" (line 283-337) is very theoretical, its content corresponds to methodology

Answer: There are several approaches to performing a correspondence analysis, but all have the same objective - to reduce samples and variables to the same metric. A fundamental step is the definition of the similarity measurement. As the software was built by the authors themselves, in Matlab language, we chose to detail the structure and concepts of the calculation algorithm, making it easily understandable and accessible to those who have knowledge of mathematics and want to build their own algorithm for the treatment of other types of information.

 Separate discussion and conclusions

Answer: Done

Reviewer 3 Report

I have read the article entitled "  The Legacy of Potential Environmental Soil Contamination in an Antimony Mining Heritage Area" written by Antonio Manuel Antunes Fiuza, Aurora Magalhães Futuro da Silva , Joaquim Eduardo Sousa Gois , Maria Lurdes Proença Amorim Dinis , Maria Cristina Costa Vila , José Manuel Soutelo Soeiro Carvalho , Antonio Carlos Pinheiro Fernandes, who carry out their professional research work at CERENA—Polo FEUP—Centre for Natural Resources and the Environment, Faculty of Engineering, University of Porto, 4200-465 Porto, Portugal.

The paper is 16 pages long.

The paper is well organized and contains the following sections:  Abstract, 1 Introduction, 2 Site description, 3 Materials and Methods, 4 Data analysis, 5 Toxicity of Antimony and Arsenic, 6 Discussion and Conclusions.

All sections are well developed.

I have done a reading and review of its parts:

In order to talk about mining heritage, there should be mining facilities that justify talking about it in that sense, the authors do not justify it adequately. They do so from a historical point of view, which is fine if they had gone a step further and carried out an environmental forensic study of the area. To talk about heritage, there has to be more than a mining history of ancient exploitation. There is a lack of images of the installations, dumps, buildings and mine shafts. In any case, I would recommend using orphan mining area, as it is more in line with what the authors are studying.

In a natural geochemical anomaly caused by an ore deposit, reference levels for contaminated soils should not be used, as the amount of metals present is not of anthropogenic origin. If they are used, the reason for this should be well justified, e.g. a health risk if food is grown for human consumption.

In the introduction I think it is important to add something about the antimony problem linked to orphan or abandoned mining in order to give more weight to the article and to be able to draw better conclusions than at present. For example:

10.1016/j.envint.2021.106908

10.1155/2019/2754385

10.1007/s10653-010-9284-z

10.3390/ijerph20010242

10.1007/s11368-010-0196-4

10.1007/s10230-017-0479-8

10.1474/Epitome.04.0323.Geoitalia2011

10.1016/j.scitotenv.2014.07.117

10.1016/S0927-5215(00)80014-4

 10.1007/s13762-020-02938-z

 10.1021/es2022472

The graphical part is very well worked out in figures 1 to 6;

In figure 2, the watercourses should be added, and on the orthophotography, the outlines of the waste dumps, mine entrances and any other surface facilities should be added. I am missing in all the graphical part a photograph of the dumps and other mining installations, in order to understand the authors' discourse. Here it would also be important to delimit the plots where agriculture, forestry and low scrubland exist. Reference is made to agricultural use in line 84.

For me figure 3 and 4 should be in the same figure, one next to the other. The reference of the sampling point would need to be added to each point in the image.

In figure 8 it would be useful, for clarification, to delineate the outlines of the locations of waste dumps and other mining facilities, as well as stream beds.

Figure 10 of the article does not fit with what the authors are saying in the text, I explain: the authors say that they use Matlab to carry out the PCA and in figure 10 they show the projection of the variables on the first factorial plane and the second (F1-F2). This figure is not made in Matlab, but is from ANDAD (Sousa, J., 2002. ANDAD Program Version 7.10 - User's Manual. CVRM - Geosystems Centre. Technical University of Lisbon [in Portuguese])

In fig. 1 I show how the figure of the authors is the same as in an article where ANDAD is used to do the PCA. I have been working with Matlab for more than 20 years and I do not understand the authors' reason for making the figures in Excel when Matlab has quality graphical output, as shown in figure 2.

Fig. 1.- PCA figure 10 from author’s paper. PCA from DOI: 10.1016/j.gexplo.2010.08.001;

Fig. 2.- Matlab PCA from DOI: 10.1016/j.chemolab.2015.10.003

Ballabio, D. 2015. A MATLAB toolbox for Principal Component Analysis and unsupervised exploration of data structure. Chemometrics and Intelligent Laboratory Systems, Volume 149, Part B, 2015, Pages 1-9, ISSN 0169-7439, https://doi.org/10.1016/j.chemolab.2015.10.003.

If the analysis is done with Matlab, it is better to do all the figures with Matlab so that they are homogeneous and not a pastiche of Excel and other programmes, as it is detrimental to the quality of the article. If ANDAD is used, the figures shown by the authors on PCA and CA can be done with it and the graphical outputs are very good.

I am not sure that the method used to obtain the sample is the most appropriate (line 140), for a site like this I would have opted for a composite sample taken with a crow's foot pattern, with which four subsamples would be taken that would make up the sample for that point.

In line l44 the authors talk about calculating the porosity of the sample and the humidity, I think the humidity is correct, but the porosity of an altered sample I don't know what interest it could have.

What is the particle size of the samples? Very large particle sizes cannot be prepared in a porcelain mortar, what was done with them, are the metals that are present in them and can leach into the environment disregarded? Why line 146 talks about homogenisation of the sample, was it ground in several steps and then homogenised?

It is important to explain how the sample has been divided in order to know if the sub-samples are representative, has the theory of Pierre Gy been followed to ensure the representativeness of the sample? If so, explain how this division of the sample was made.

From line 151 to 163 all the analyses carried out are explained, where is this data shown? I don't see them in the article and more importantly I don't see them in the discussion of the article. Of all the tests that the authors say they have carried out, they only show a summary table of the XRF analyses but they do not say what fraction it is, remember that in line 156 they say that they carry out two XRF analyses, one on the overall sample and the other on the sample smaller than 0.075 mm. What fraction are the data in Table 1 from?

For this initial part I am missing a violin plot of the data, or alternatively a boxplot, although it provides less information.

In lines 186- 187 it is said that "... This methodology applied to the samples of both populations evidenced that the populations have different means at both levels of significance..." where the data that justifies it is shown, taking into account that the authors do not give us the data and we cannot check it for ourselves and we have to believe what they say, what data supports such a categorical statement?

The same applies to lines 204-206.

In section 5 (lines 348-360) the toxicity of antimony and arsenic is discussed, but I find the section very poor in its discussion, it lacks a risk analysis of the areas, especially those where there may be agricultural uses with plantations destined for human consumption.

For lines 362-365 I refer to what has been said above, there is no environmental or geochemical justification for using contaminated land legislation for an area with a geochemical anomaly caused by an antimony-gold mining deposit, unless the authors justify it very well. Otherwise it is a misconception, as the geochemical anomaly defining the ore deposit will exceed the values of the contaminated land legislation, hence the term geochemical anomaly.

The authors should consider adding a table of data, either in the article or as supplementary material to give weight to their work, as it is presented as it is. The other option would be to publish these data in DATA.

I encourage the authors to continue working in this area and expand the studies towards leachate generation, bioavailability and environmental forensic studies.

Round 2

Reviewer 1 Report

I'm satisfied with these additions to the text. Thank you.

Author Response

No new comments. Thank you for your suggestions.

Reviewer 2 Report

The manuscript has been improved and can be published but minor changes are required.

The abstract is too long.

The quality of fig. 4 and 7 should be improved; the text is too small, the legend is not readable, there is overlapping information, etc. 

line 56-57. It is recommended to review the following article. https://doi.org/10.1007/s10661-022-10433-w

Author Response

Thank you for your suggestions. Please see my answers in the attached file.
